# Confinement of a Styryl Dye into Nanoporous Aluminophosphates: Channels vs. Cavities

**DOI:** 10.3390/ijms25073577

**Published:** 2024-03-22

**Authors:** Ainhoa Oliden-Sánchez, Rebeca Sola-Llano, Joaquín Pérez-Pariente, Luis Gómez-Hortigüela, Virginia Martínez-Martínez

**Affiliations:** 1Departamento de Química Física, Universidad del País Vasco (UPV/EHU), Apartado 644, 48080 Bilbao, Spain; ainhoa.oliden@ehu.eus (A.O.-S.); rebeca.sola@ehu.eus (R.S.-L.); 2Instituto de Catálisis y Petroleoquímica (CSIC), c/Marie Curie 2, Cantoblanco, 28049 Madrid, Spain; jperez@icp.csic.es

**Keywords:** styryl dyes, aluminophosphates, photoactive hybrid systems, MgAPO, AEL framework, CHA framework, fluorescence, nanochannels, cavities, molecular sieves

## Abstract

Styryl dyes are generally poor fluorescent molecules inherited from their flexible molecular structures. However, their emissive properties can be boosted by restricting their molecular motions. A tight confinement into inorganic molecular sieves is a good strategy to yield highly fluorescent hybrid systems. In this work, we compare the confinement effect of two Mg-aluminophosphate zeotypes with distinct pore systems (the AEL framework, a one-dimensional channeled structure with elliptical pores of 6.5 Å × 4.0 Å, and the CHA framework, composed of large cavities of 6.7 Å × 10.0 Å connected by eight-ring narrower windows) for the encapsulation of 4-DASPI styryl dye (trans-4-[4-(Dimethylamino)styryl]-1-methylpyridinium iodide). The resultant hybrid systems display significantly improved photophysical features compared to 4-DASPI in solution as a result of tight confinement in both host inorganic frameworks. Molecular simulations reveal a tighter confinement of 4-DASPI in the elliptical channels of AEL, explaining its excellent photophysical properties. On the other hand, a singular arrangement of 4-DASPI dye is found when confined within the cavity-based CHA framework, where the 4-DASPI molecule spans along two adjacent cavities, with each aromatic ring sitting on these adjacent cavities and the polymethine chain residing within the narrower eight-ring window. However, despite the singularity of this host–guest arrangement, it provides less tight confinement for 4-DASPI than AEL, resulting in a slightly lower quantum yield.

## 1. Introduction

Styryl dyes consist of an electron donor (D) and electron acceptor (A) building block linked together through a π-conjugated system, forming D-π-A architectures [1,2]. In general, they present high absorption capacity in the blue to green region of the visible spectrum and highly red-shifted fluorescence, as the emission takes place from an intramolecular charge-transfer state (ICT), exhibiting rather large Stokes shifts, an important feature to minimize inner-filter effects (reabsorption and re-emission) [3,4,5,6]. Their strong push–pull character makes them very promising organic molecules for advanced optical applications, including nonlinear optics (NLO), sensing, and bioimaging [3,6,7,8,9,10,11,12]. However, they are very flexible molecules and require very rigid environments to restrict their molecular motion in order to achieve high fluorescence or lasing efficiency; indeed, rotational motions around the different bonds are responsible for the non-radiative deactivation, such as trans–cis isomerization, or especially the formation of twisted intramolecular charge-transfer (TICT) states, which are usually non-fluorescent [13,14,15].

In this context, confining these molecules to molecular sieves could restrict their molecular motions, limiting the internal conversion. Thus, by forcing a rigid planar D-π-A molecule, an efficient ICT can take place without the involvement of large amplitude motions, yielding high fluorescence or even laser efficiency [16,17,18]. Furthermore, a preferred orientation within 1D nanochannels can lead to a noncentrosymmetric arrangement and thus to anisotropic behavior, which is crucial for nonlinear optical applications [19,20]. Therefore, it is essential to provide tight confinement for styryl dyes in order to enhance their photophysical properties and boost new features. 

In previous works, we demonstrated that the optimum confinement of LDS-722 styryl dye into the 10-ring elliptical channels of the AEL Mg-doped aluminophosphate framework [21] rendered a material with very promising photophysical properties. By simply changing the inorganic framework, shifting from the large-pore 12-ring channeled AFI framework with cylindrical channels of 7.3 Å in diameter to the medium-pore 10-ring elliptical channels of 6.5 Å × 4 Å of the AEL framework, we managed to notably improve the quantum yield of the LDS-722 styryl dye [20]. Consistently, we have on multiple occasions shown that a proper size match between the dye and the host channel dimensions is crucial for achieving efficient hybrid photoactive systems [22,23,24,25]. 

We subsequently considered the possibility of achieving an even tighter confinement of styryl dyes to further improve their photophysical properties. Nevertheless, a further reduction of the pore size from 10 rings as in AEL to 8 rings (small pore) might prevent the occlusion of these styryl dyes due to the relatively bulky end groups (N, N-dimethylaniline and pyridinium rings) that would prevent their allocation within such small channels. Thus, apparently, the tightest confinement for these styryl-type dye molecules employing channel-based zeolites would have been reached. However, other types of frameworks exist among zeolite materials whose void volume is constituted by large cavities that are connected by smaller windows [26,27,28,29]. These cavity-based zeolite systems provide new potential hosts for styryl-type dyes, which could be eventually confined within these particular void spaces, imposing significant restrictions on the motion of the photoactive molecule. This type of confinement on cavities is singular since it limits the motion of guest species in all three directions, unlike that occurring within 1D channels where motion along the channel direction is free. Consequently, the restricted motion within the zeolite cavities holds promise for enhancing the fluorescence properties of guest dyes confined within.

In this family of zeolite frameworks based on cavities, the chabazite structure (CHA structure code) is one of the most widely known and used [30]. The CHA structure is a three-dimensional system, with cages accessible through eight-membered ring windows with 3.8 Å diameter, opening into large 6.7 Å × 10.0 Å ellipsoidal cavities (a schematic representation is shown in Figure 1c) [31]. Hence, in order to analyze the effect of this new type of confinement of dye molecules within zeolite cavities, in this work we study and compare the photophysical properties of a styryl dye with appropriate size, 4-DASPI (trans-4-[4-(Dimethylamino)styryl]-1-methylpyridinium iodide), occluded within two types of confinements: within the one-dimensional medium-size pores of AEL channels and within the cavities of the CHA framework (Figure 1). The incorporation of dye molecules into the pores and channels of zeolites is achieved through the crystallization inclusion method [32]; notably, the application of this method to host photoactive dye species within the CHA zeolite framework represents a novel approach, as previous studies with dyes have primarily relied on diffusion methods [33,34]. In fact, this is the only plausible method that allows the occlusion of these particular types of dyes, as the diffusion process would be completely impeded by the bulky aromatic end groups and the small-ring windows accessing the cavities in the CHA framework.

## 2. Results and Discussion

### 2.1. Synthesis of 4-DASPI@MgAPO-CHA System

A dye-free synthesis was first performed to confirm the formation of pure MgAPO-CHA (MgAPO-34 material) under the specified conditions (see Section 3.1). Secondly, the dye was incorporated into the synthesis gel to obtain the dye-loaded 4-DASPI@MgAPO-CHA hybrid system. 

Powder X-ray diffraction (PXRD) data (Figure 2) showed the crystallization of pure CHA materials in both cases (XRD peaks coincided with those reported for the pure CHA framework [21]), the crystallinity being slightly lower when the material was prepared in the presence of 4-DASPI, suggesting that the dye slightly hindered the crystallization of CHA. The 4-DASPI@MgAPO-CHA solid showed a light orange color, indicating a limited presence of the dye in the structure. The estimated amount of encapsulated dye was relatively low, around 0.3 mmol per 100 g of powder sample.

### 2.2. Synthesis of 4-DASPI@MgAPO-AEL System

Initially, the same conditions already used in our previous works were set in order to promote the crystallization of MgAPO-AEL in the presence of 4-DASPI (see Table 1, entries 1 to 4) [20]. However, PXRD patterns showed the concomitant crystallization of the AFI phase in all the conditions tested (Figure 3, dashed lines) along with AEL, which had not been observed in the absence of the dye. These results clearly indicate that, as also occurred for LDS 722, 4-DASPI acts as an efficient structure-directing agent for the large-pore AFI framework, even if its amount with respect to the organic cation used as the template (ethyl-butylamine, EBA) is very small (0.024 dye vs. 1.00 EBA). In fact, by using the same gel molar composition with many other dyes (e.g., acridine, methylacridine, pyronin Y), a pure AEL phase was always obtained without difficulty [24,25]. A rough estimation of the relative presence of AEL and AFI phases (by comparing the ratio of the peak intensities corresponding to each phase) showed that the AFI phase was favored at high EBA concentration, i.e., at higher pH values (pH ≈ 5). Thus, in an attempt to minimize the amount of AFI crystallized, we kept a low EBA content (0.75) and reduced the amount of dye (0.008) to minimize its structure-directing effect towards AFI. Furthermore, we tried two different amounts of phosphoric acid (1.00 and 1.20 P_2_O_5_, entries 5 and 6 in Table 1) to keep the pH low (pH ≈ 4). Interestingly, these new synthesis conditions allowed us to further reduce the AFI concomitant phase to a minimum (entry 6, see Figure 3), although it was not completely prevented (Table 1).

### 2.3. Photophysics of 4-DASPI Dye in Solution

In order to better understand the photophysical properties of the host–guest systems, studies of the fluorescence properties of 4-DASPI molecules in aqueous media are a prerequisite. Table 2 shows the main photophysical parameters of 4-DASPI fluorophore in aqueous medium. Briefly, it absorbs in the blue region of the electromagnetic spectrum (λ_ab_ = 449.5 nm) and emits in the red (λ_fl_ = 618.5 nm, Appendix A). As mentioned above, the large Stokes shift (∆ν_st_ = 6079 cm^−1^) is due to the push–pull nature of this type of dyes, where fluorescence comes from an intramolecular charge transfer between the donor (dimethylaniline) and acceptor (pyridinium) moieties favored in the excited state (see the difference in the electronic distribution between HOMO and LUMO orbitals in Appendix A). The dye shows poor fluorescence in solution (ϕ_fl_ < 0.01 and τ < 0.1 [35]), the molecular motions being responsible for the non-radiative deactivation as the main pathway to the ground state [36,37] (Figure 1b shows the main rotation of 4-DASPI around the single bond between the aniline ring and the pyridylethylene motifs).

As mentioned in the introduction, since these photophysical parameters depend on both the polarity and the viscosity/rigidity of the medium, they can be modulated and significantly improved by freezing this dye within a rigid solid matrix with narrow channels [38]. Consequently, the goal is mainly to limit the flexibility of the molecule by slowing down the molecular motions and TICT in the excited state.

### 2.4. Photophysics of 4-DASPI@MgAPO-CHA

With regard to the synthesized hybrid materials with the CHA framework, Figure 4 shows the occurrence of small MgAPO-34 crystals, with an average dimension of about 1–3 µm. Polarized fluorescence experiments showed that 4-DASPI molecules in MgAPO-34 crystals do not show a preferential orientation (dichroic ratio, D = I∥/I⊥, defined as intensities recorded for orthogonal emission polarizations, is close to 1). As expected, when the analyzer (set before detection) was systematically tuned from 0° to 90° in several steps, the fluorescence signal of particles did not show any intensity dependence, which indicates isotropic behavior (no preferential orientation) of 4-DASPI molecules as a consequence of their 3D disposition within this type of framework, where they are incorporated inside the cages.

The photophysical characterization of the 4-DASPI@MgAPO-CHA solid sample (Figure 5 and Table 2) reveals two broad absorption bands centered at 343.0 and 437.0 nm. These bands are assigned to the presence of 4-DASPI dye in different protonation states, dicationic (DC) and monocationic (MC) species (Appendix A), respectively, as confirmed by the photophysics of the dye in solution at different pHs. This indicates that 4-DASPI is incorporated in both protonation states within the CHA framework (Figure 5a). In contrast, the fluorescence band centered at 583.0 nm is assigned to the CT state of the MC species, the emission of the DC species being practically negligible (both in solution and in CHA). This band is blue-shifted with respect to 4-DASPI in aqueous solution (618.5 nm), a typical observation for chromophores encapsulated in very rigid media. The high Stokes shift (∆ν_st_ = 5416 cm^−1^) observed when confined in this CHA framework, although slightly lower than in solution (∆ν_st_ = 6079 cm^−1^), together with the bathochromically shifted band registered in the excitation spectrum (maximum at 525 nm, Figure 5b) with respect to its original absorption at 437.0 nm, confirms the charge-transfer nature from MC as being primarily responsible for the fluorescence emission.

It is worth noting at this point that the 4-DASPI dye loaded into the CHA material is highly fluorescent regardless of the excitation wavelength, reaching a quantum yield of around 33% (over 30 times higher than in solution), leading also to a greatly enlarged decay lifetime of 2.7 ns, much longer than those obtained in solution (<0.1 ns) (Table 2), indicating the substantial confinement and motion restriction imposed by the CHA framework on the occluded dye.

### 2.5. Photophysics of 4-DASPI@MgAPO-AEL

Remarkably, a large amount of dye is occluded into the AEL-rich samples, giving values from 2 to 9 mmol per 100 g of sample powder (Table 1), which represent a percentage from 25 to almost 95% of the dye initially added to the gel, resulting in a very intense orange-reddish powder (Appendix A), rendering very broad absorption bands (Appendix A). The broadening of the absorption band is a consequence of the great stabilization of the CT state in the ground state, overlapping with the absorption band of the locally excited “LE” state of 4-DASPI. Note here that the presence of the AFI phase affects the final dye-uptake values, overestimating the amount occluded in AEL, since the higher the presence of AFI, the higher the loading of the 4-DASPI dye obtained (note that quantification is performed on the powder bulk sample, where it is not possible to distinguish between phases; see Materials and Methods section) (Table 1). This higher incorporation of 4-DASPI in the AFI framework may be closely related to the dimensions of the unidirectional channels of the framework, where the wider and more cylindrical channels of AFI (7.3 Å) facilitate the incorporation of 4-DASPI molecules compared with the elliptical and narrower channels of AEL (4.0 Å × 6.5 Å), where confinement is tighter. The presence of AFI impurities not only affects the dye loading but also the overall emission properties of the samples. For instance, sample 1, characterized by the highest amount of AFI, exhibits an overall fluorescence yield of 10%, while this value increases to 41% in sample 6, with the lowest presence of this undesirable phase (Table 2), despite showing lower dye uptake. This fact is again related to the larger pores of the AFI structure, whose dimensions are too wide to boost the photophysical properties of the dye. This statement is confirmed by fluorescence microscopy. Figure 6a,b shows large crystals (>20 μm) with the characteristic well-defined hexagonal and elongated particles typical of the AFI phase, which are intergrown with rectangular plate-like particles characteristic of the AEL phase. AFI crystals are not fluorescent, whereas AEL crystals display high and homogeneously distributed fluorescence. Furthermore, the one-dimensional channels of the MgAPO-AEL framework ensure a preferred alignment of the organic dye molecules along the pores, as observed through fluorescence microscopy, where large dichroic ratios (D values of around 30–40) were determined for the 4-DASPI@MgAPO-AEL single particles (Figure 6c,d).

4-DASPI@MgAPO-AEL samples show an emission band centered at around 610.0 nm, which is slightly blue-shifted with respect to solution (618.5 nm). The smaller blue shift registered for these samples with respect to that found in the fluorescence band for 4-DASPI@MgAPO-CHA (583.0 nm) could point to a less tight confinement. However, the higher emission efficiency (ϕfl = 41%) and longer lifetimes recorded for 4-DASPI@MgAPO-AEL-6 (τ = 3.3 ns) with respect to 4-DASPI@MgAPO-CHA (2.7 ns) indicates the opposite, since larger fluorescence quantum yields and lifetimes are a consequence of a reduction in non-radiative processes as a result of higher restrictions for molecular movement (this statement is further supported in Section 2.6). Hence, the observed variations in the emission band positions for 4-DASPI into AEL and CHA can be attributed to two possible factors: (i) distinct geometries of the excited charge-transfer (CT) state, as will be explained below (Figure 7), where the dye molecular structure confined within the CHA cavities shows a larger deviation from planarity (Figure 7a) compared to that within AEL (Figure 7b), leading to a larger blue shift and a less energy-stabilized CT state [39]; and (ii) the inner filter effect, where the greater dye uptake and higher overlap between absorption and emission bands in 4-DASPI@MgAPO-AEL samples (Appendix A) contribute to a red shift in emission. Thus, the AEL matrix seems to impose a stronger restriction on 4-DASPI dye molecules, decreasing non-radiative pathways more efficiently.

Note here that for sample 4-DASPI@MgAPO-AEL-1, the lower fluorescence quantum yield and the biexponential behavior observed in the fluorescence decay curves (Table 2) are once again attributed to the presence of AFI impurities. This results in a less emissive system and, consequently, shorter lifetime values. Hence, the longest lifetime of about 3 ns with a weight of 80% is ascribed to 4-DASPI dye within the AEL structure, while the shortest lifetime of about 0.13 ns with a weight of 20% indicates a less rigid environment, which is likely due to 4-DASPI within the AFI structure.

In summary, the photophysics of the 4-DASPI dye, in particular the emission efficiency and lifetimes, have been significantly improved with respect to 4-DASPI in solution by the encapsulation of this dye within the cavities and channels of CHA and AEL zeolitic structures, respectively, with only the latter displaying a preferential orientation of the molecules. Nevertheless, the confinement seems to be tighter in the AEL framework, giving higher fluorescence. The results obtained for 4-DASPI within the channels of AEL are similar to those previously derived from the encapsulation of another styryl dye, LDS 722, into AEL, in which a sample with similar dye uptake rendered a fluorescence quantum yield of 41% and lifetime of 2.5 ns [20], demonstrating that this type of framework with 1D elliptical channels of 6.5 Å × 4 Å is a perfect choice for the confinement of this family of dyes. Furthermore, styryl@AEL has a higher dye loading and a preferential orientation of the dye molecules along 1D channels, which could potentially trigger intriguing NLO optical properties such as SHG, as has been formerly demonstrated for the LDS 722@AEL system [20]. Further experiments are being planned to explore optical properties such as second-harmonic generation (SHG) and laser action. For that, additional synthesis will be conducted to attain particles with suitable size, shape, and dye loading.

### 2.6. Molecular Simulations

We finally analyzed the confinement of 4-DASPI dyes (in MC protonation state, which is predominant) within the two frameworks by molecular simulations (based on the cvff forcefield). Figure 7 displays the most stable configuration for the MC species of 4-DASPI dye in the CHA (a) and AEL (b) frameworks. Energy results indicate a much higher stability (more negative interaction energy, I.E.) for 4-DASPI confined within the AEL framework (−94.0 compared to −52.9 for CHA), thus explaining the higher incorporation experimentally found in AEL. As for the related LDS 722 system, 4-DASPI molecules are well aligned within the AEL channels (Figure 7b), showing a very good host–guest fit that imposes a high restriction on motions and deviation from planarity [20]. A different configuration is found for 4-DASPI confined within the CHA framework; CHA cavities were found to be not large enough as to host a complete 4-DASPI molecule; hence, molecular simulations suggest that the dye would locate along adjacent cavities, siting the poly(methine) intermediate chain within the eight-ring windows that connect the two cavities, and each bulky aromatic group within each cavity (Figure 7a). This type of configuration has been previously found for organic cations acting as structure-directing agents with a similar molecular structure as 4-DASPI based on bulky aromatic rings linked by alkyl chains [40]. Interestingly, this particular configuration involves 4-DASPI dye molecules being oriented in any of the three directions perpendicular to the eight-ring windows connecting the CHA cavities, as can be observed in Figure 7a (inset), which explains the lack of a preferential orientation (dichroic ratio) found experimentally.

In order to estimate the restrictions on motion imposed by each framework, 500 ps MD simulations were run (at 298 K). Deviation from planarity between the two aromatic rings was estimated by plotting the distribution of the torsion angle between the two rings (Figure 7c), while rotation around the aniline C-N bond was estimated by the distribution of the corresponding torsion angle along the MD simulation (Figure 7d). MD results showed a higher restriction to deviation from planarity of the aromatic rings and to rotation of the C-N bond for the AEL framework (estimated as the higher density of probability for torsion angles of 0 deg), hence suggesting that the AEL framework imposes a higher confinement and restriction to motion to 4-DASPI, thus explaining the experimental observations of the higher quantum efficiency and longer lifetime values for 4-DASPI@AEL.

**Figure 7 ijms-25-03577-f007:**
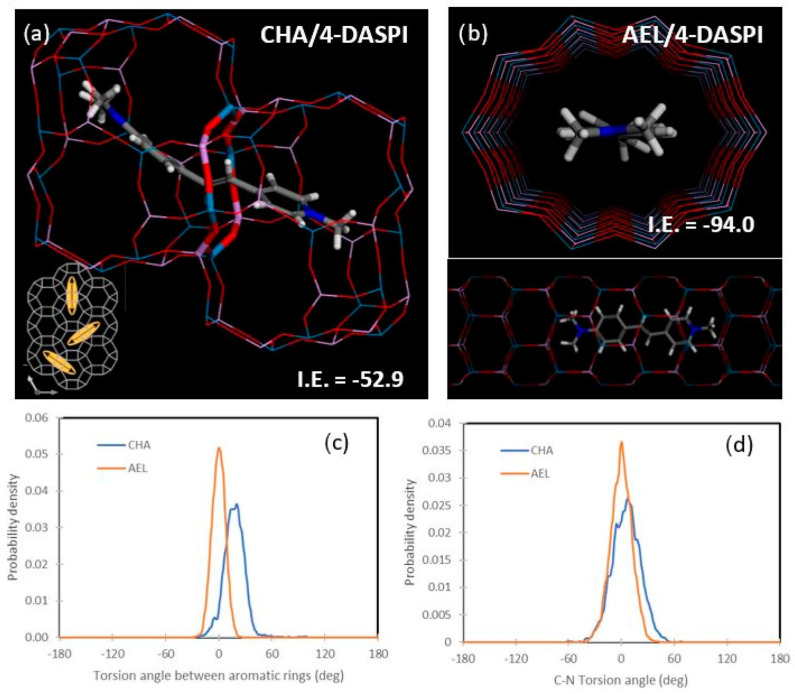
Molecular simulations of 4-DASPI confined within CHA or AEL frameworks. Most stable location within CHA (**a**) or AEL (**b**) frameworks and distribution of torsion angle between the two aromatic rings (**c**) and of the C-N (aniline) bonds (**d**).

In contrast, in the case of 4-DASPI confined within the cavity-based CHA framework, siting the intermediate chain within a small eight-ring window would provide a tight confinement to the (poly)methine chain, albeit at the expense of locating the bulky aromatic ring groups within the large cavities that would impose less restriction to motion, resulting in slightly less restricted confinement, where dyes do not display a preferential orientation. This can be considered a novel hybrid system with a rather particular dye arrangement, giving rise to distinctive photophysical properties.

## 3. Materials and Methods

### 3.1. Synthesis of Hybrid Materials

Typical syntheses of MgAPO-CHA involve high temperature of 190 °C [41], which may cause degradation of the dye. For this reason, we set up a new synthetic protocol at a lower crystallization temperature (150 °C), which involved the addition of a higher amount of TEA as SDA and magnesium. Hence, synthesis of MgAPO-CHA was performed from gels with the following molar composition: 3.3 TEA: 0.47 MgO: 0.765 Al_2_O_3_: 1 P_2_O_5_: 0.024 4-DASPI: 47 H_2_O. In a typical synthesis procedure, pseudoboehmite (Al_2_O_3_, Sasol, Johannesburg, South Africa, 75%) was added to a solution of orthophosphoric acid (H_3_PO_4_, Aldrich, Darmstadt, Germany, 85%) in water and was stirred for 15 min. Next, triethylamine (TEA, Aldrich, 99.5%) was added to the solution and stirred for 10 min, followed by the addition of magnesium acetate tetrahydrate (Mg(acet)_2_·4 H_2_O, Aldrich, 99%) and an additional 10 min of stirring. Then, 4-DASPI (trans-4-[4-(dimethylamino)styryl]-1-methyl-pyridinium iodide, Aldrich, 98%) was added, and stirring continued for 2 h [41]. The gels were introduced into autoclaves to be heated statically in a conventional oven at 150 °C for 48 h.

In the synthesis of 4-DASPI@MgAPO-AEL, the initial molar composition of the gel was the following: 0.95 Al_2_O_3_: 1 P_2_O_5_: x MgO: y EBA: z Dye: 300 H_2_O. The same was used in our previous works [20]. However, since we found the co-crystallization of the AFI phase as a consequence of the presence of 4-DASPI, we slightly varied x, y, and z in order to minimize the presence of such competing phase; details are given in the Results and Discussion section. In a general procedure, orthophosphoric acid was mixed with Milli-Q water and kept under vigorous stirring for 2 min. Next, magnesium acetate tetrahydrate and aluminum hydroxide (Al(OH)_3_, Aldrich) were gently added, and the resulting mixture was left under stirring for 10 min. Ethyl-butylamine (EBA) was then added to the reaction gel together with the organic dye, and stirring continued for 1 h. The aqueous mixture was then heated statically at 180 °C in an autoclave for 24 h in a conventional oven. All solid products were recovered by filtration, exhaustively washed with ethanol and water, and dried at room temperature overnight.

### 3.2. Characterization Techniques

Powder X-ray diffraction (PXRD) was used to identify the crystalline framework structures of the solids, by comparison with the database of zeolite structures [21]. The diffraction patterns were collected by using a Philips X’pert PRO automatic diffractometer (Malvern Panalytical, Lelyweg, The Netherlands) with Cu-Kα radiation (λ = 1.5418 Å) and a PIXcel solid-state detector (Malvern Panalytical, Lelyweg, The Netherlands). The size and morphology of the samples were characterized by SEM in a JSM-6400 (tungsten filament) operating at 20 kV and 10^−11^ A (JEOL, Tokio, Japan). These measurements were carried out at the University of the Basque Country UPV-EHU, SGIker facilities.

### 3.3. Quantification of DASPI Loading into MgCHA and MgAEL

The final dye loadings incorporated into the aluminophosphate powders were quantified spectrometrically after dissolving a certain amount (10 mg) of the sample powder in hydrochloric acid (5 M). The absorption spectra of the samples were compared with the absorptions of the previously prepared standard solutions (with known concentrations), making a calibration curve at analogous conditions. The absorption spectra were recorded with a UV-Vis spectrophotometer (Agilent technologies Cary 7000, Madrid, Spain), described below. Dye content values are given throughout this work as mmol of dye per 100 g of sample powder and in percentage with respect to the initial amount added in the synthesis gel.

### 3.4. Photophysical Characterization

Absorption spectra were recorded in a double beam Cary 7000 spectrophotometer (Agilent technologies, Madrid, Spain) with a Hamamatsu R928 photomultiplier as a detector, in transmittance for solutions (organic dye, standards, and dye-loading quantification), by baseline correction, and with an integrating sphere for bulk powder (Internal DRA 900, Agilent technologies, Madrid, Spain). Steady-state fluorescence measurements were carried out in a FLS920 Spectrofluorometer (Edinburgh Instruments, Livingston. U.K.), with a photomultichannel tube (PMT) (Hamamatsu R2658P) as a detector. Fluorescence quantum yields for powder samples were determined via absolute method, through an integrating sphere mounted into FLS 920 spectrofluorometer; radiative decay curves were recorded in the same spectrofluorometer with a time-correlated single-photon counting technique using a microchannel plate detector (Hamamatsu C4878) with picosecond time resolution. The fluorescence lifetimes (τ_fl_) were obtained after deconvolution of the instrumental response signal from the recorded decay curves by means of an iterative method. The goodness of the exponential fit was controlled by statistical parameters (chi-square χ^2^ values between 0.9 and 1.3 and analysis of the residuals). Usually, in solid samples, due to their heterogeneity in comparison to the solution, multiexponential behavior can be observed. Thereupon, the decay curves are normally adjusted to a sum of exponential decays using the equation: I_fl_ (t) = Σ A_i_ e^(t/τi)^, where A_i_ is the pre-exponential term, and τ_i_ is the fluorescence lifetime. Analysis done by F900 software (version 7.3.4, copyright@2012, Edinburgh Instruments, Livingston. U.K). Fluorescence microscopy images were recorded with an optical upright wide-field microscope with epi configuration (Olympus BX51, Evident, Hamburg, Germany,) equipped with a color CCD camera (DP72, Evident, Hamburg, Germany,) using HQ530/30m band pass, Q660LP dichroic filter, and E580lp cut-off (all from Chroma Technology, Rockingham, USA). Dichroic ratio D, defined as the ratio between the intensities recorded for orthogonal emission polarizations, is determined as an indicative parameter of molecule orientation within the pores as: D = I_∥_/I_⊥_, where I_∥_ and I_⊥_ are the fluorescence intensities measured with the analyzer parallel and perpendicular to the sample crystal axis, respectively.

### 3.5. Computational Details

Ground state (S_0_) geometries of isolated 4-DASPI were optimized by Density Functional Theory (DFT) using the B3LYP hybrid functional and the triple valence basis set with one polarization function (6–311 g*) [42,43]. The geometries were checked by frequency analysis to know whether they corresponded to a true energy minimum, and this was confirmed when the analysis did not give any negative values. From the optimized geometries, the molecular dipole moments of the organic dyes and the molecular orbitals HOMO (Highest Occupied Molecular Orbital) and LUMO (Lowest Unoccupied Molecular Orbital) were determined. All the simulations were conducted at the Gaussian 16 software in the computational cluster ARINA of the University of the Basque Country UPV-EHU.

Calculations on the location of dyes embedded within the CHA and AEL framework structures were based on molecular mechanics, as implemented in Forcite module, in Materials Studio software (Materials Studio 2022, BIOVIA, San Diego, CA, USA). Molecular mechanics calculations were based on the cvff forcefield [39,44]. The most stable location of DASPI cations within the two frameworks was obtained by first loading one dye molecule by Monte Carlo simulations (Sorption module with fixed loading at 1) and then obtaining the most stable configuration by simulated annealing calculations. NVT Molecular Dynamics simulations (Forcite module) were run at 298 K in order to analyze the motion of 4-DASPI within the two frameworks, calculating the characteristic torsion angle distribution along the MD simulations.

## 4. Conclusions

By employing various magnesium-doped aluminophosphates (AEL and CHA) to provide a rigid environment for hosting 4-DASPI dye molecules, fluorescence quantum yields and excited-state lifetimes have been significantly increased by hindering the torsional motion of the dye. The different inorganic matrices influence and modulate the photophysical properties of this fluorophore, offering different functionalities to the resulting hybrid material. Thus, depending on the type of framework system, based on channels or cavities, highly fluorescent systems either with a preferential dye alignment or smaller crystals with isotropic response have been obtained for potential applications such as SHG (Second Harmonic Generation) or laser action devices, respectively. The novel design of the Dye@MgAPO-CHA system based on cavities, in which dye molecules are hosted by occupying large cavities connected by smaller windows where the polymethine chain resides, opens up new possibilities to explore other hybrid systems, which are likely to provide new optical functionalities.

## Figures and Tables

**Figure 1 ijms-25-03577-f001:**
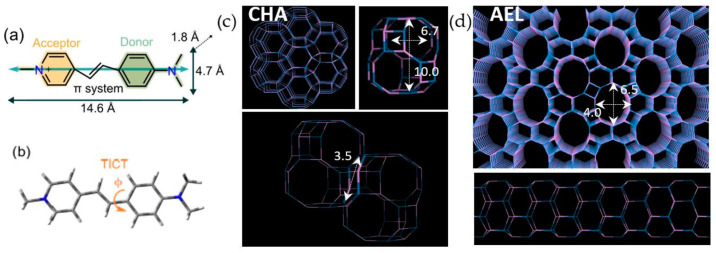
(**a**) Molecular structure (indicating donor and acceptor groups), dimensions, and transition dipole moment (blue arrow, Mulliken) of 4-DASPI dye estimated by theoretical calculations; (**b**) description of internal twisting through φ as the main deactivation process of electronically excited 4-DASPI [13]. (**c**) Three-dimensional cavities of MgAPO-34 (CHA) framework, showing the connection between adjacent cavities (down) and (**d**) 1D channels of MgAPO-11 (AEL) framework.

**Figure 2 ijms-25-03577-f002:**
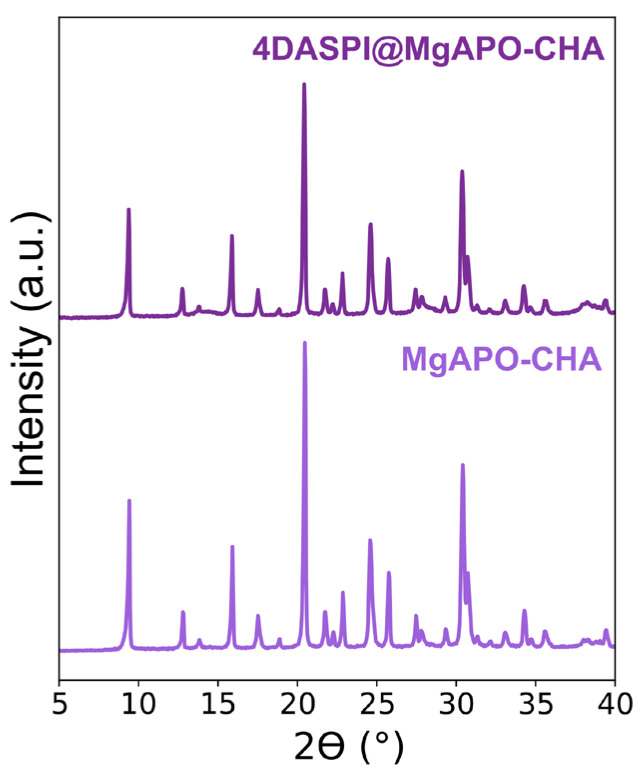
PXRD patterns of CHA samples prepared without (MgAPO-CHA) and with 4-DASPI dye (4-DASPI@MgAPO-CHA).

**Figure 3 ijms-25-03577-f003:**
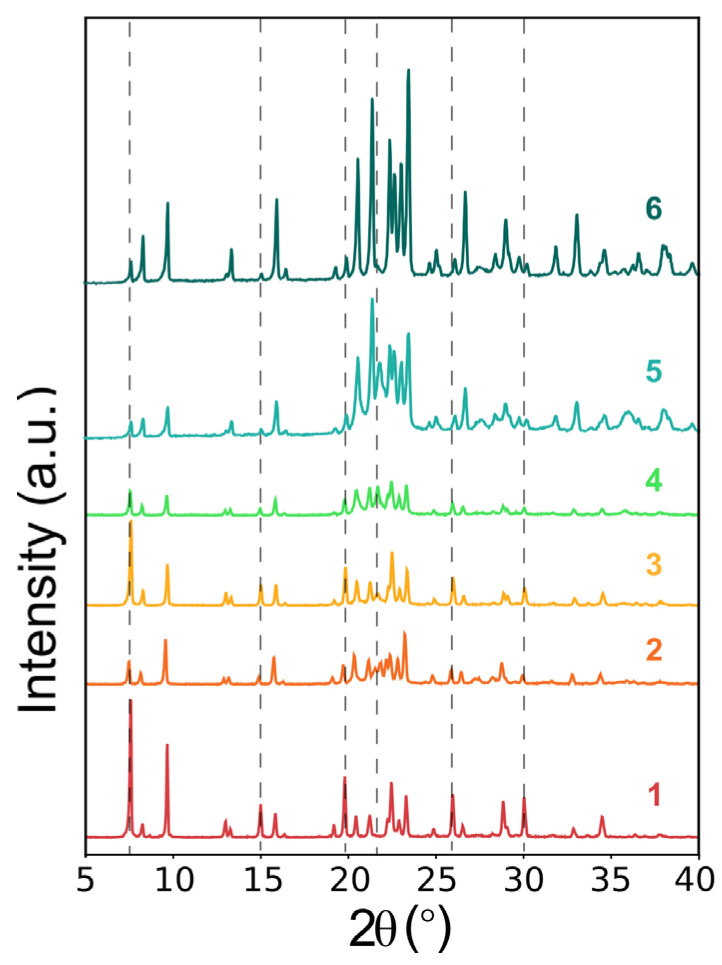
PXRD patterns of 4-DASPI@MgAPO-AEL hybrid systems (samples names according to Table 1 entries). Dashed lines denote AFI peaks.

**Figure 4 ijms-25-03577-f004:**
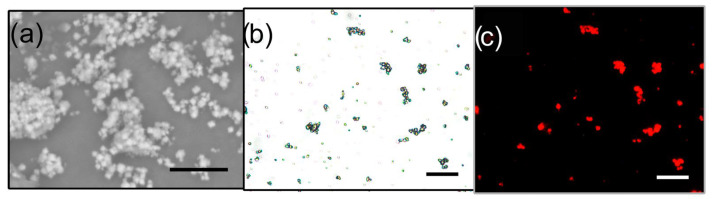
Images taken for 4-DASPI@MgAPO-CHA sample by (**a**) scanning electron microscopy (SEM) and optical microscopy: (**b**) transmission and (**c**) fluorescence (taken under green light excitation, HQ530/30 m bandpass) images. The scale is 20 μm in all of them (black and white bars).

**Figure 5 ijms-25-03577-f005:**
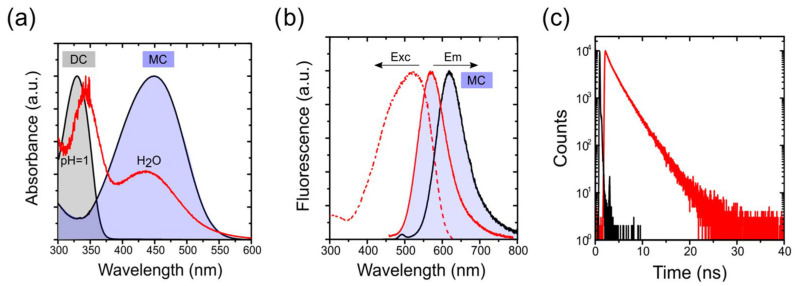
(**a**) Height normalized absorption spectra for 4-DASPI@MgAPO-CHA sample (red) and 4-DASPI in aqueous solution (black) (absorption band of diprotonated 4-DASPI at acidic pH = 1 is also included), (**b**) height normalized excitation (dotted-red) and fluorescence for 4-DASPI@MgAPO-CHA sample (solid-red) and 4-DASPI in aqueous solution (black) at λ_exc_ = 450 nm and λ_exc_ = 420 nm, respectively, and (**c**) fluorescence decay curves recorded for 4-DASPI in solution (black) and for 4-DASPI@MgAPO-CHA powder sample (red) at λ_ex_ = 450 nm and recorded at their respective emission maxima.

**Figure 6 ijms-25-03577-f006:**
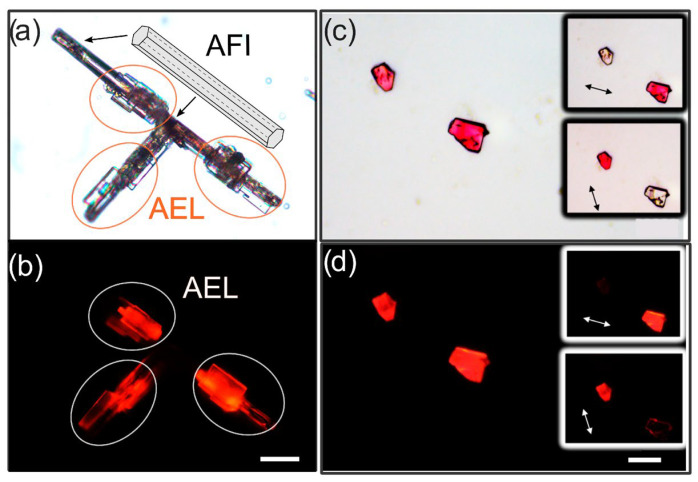
(**a**–**c**) Transmission and (**b**–**d**) fluorescence images under green light excitation (HQ530/30 m bandpass) taken for sample 4-DASPI@MgAPO-AEL-1 (**left**) and 4-DASPI@MgAPOAEL-6 (**right**). Note: an illustrative scheme of AFI–type crystals is included in the transmission image. Polarized transmission and fluorescence images are inset (arrows indicate the direction of the polarized light); scale bar = 20 μm.

**Table 1 ijms-25-03577-t001:** Variations in gel composition, final dye uptake (expressed as mmol of dye per 100 g of solid product and as percentage of the dye loaded with respect to the initial amount in the gel), and the final phases achieved (arrows indicate qualitatively the quantity of each phase).

Sample 4-DASPI@MgAPO-AEL	Gel Composition				Dye Uptake		X-ray
	x (MgO)	y (EBA)	P_2_O_5_	z (DASPI)	mmol/100 g	%	Phase
1	0.20	1.00	1.00	0.024	8.8	93	AFI(↑) + AEL(↑)
2	0.20	0.75	1.00	0.024	3.1	27	AEL(↑) + AFI(↓)
3	0.10	1.00	1.00	0.024	4.8	41	AFI(↑) + AEL(↑)
4	0.10	0.75	1.00	0.024	3.8	26	AEL(↑) + AFI(↓)
5	0.20	0.75	1.00	0.008	1.7 ^1^	26	AEL(↑) + AFI(↓↓) + try
6	0.20	0.75	1.20	0.008	2.3	34	AEL(↑) + AFI(↓↓↓)

^1^ The lower dye uptake is due to the presence of the dense, non-porous tridymite (try) in the sample.

**Table 2 ijms-25-03577-t002:** Photophysical parameters: absorption (λ_ab_), excitation (λ_ex_), and fluorescence (λ_fl_) maximum wavelengths, lifetimes (τ), and fluorescence quantum yields (ϕ_fl_) of 4-DASPI fluorophore in different environments (H_2_O, MgAPO-CHA and MgAPO-AEL).

4-DASPI@	λ_ab_ (nm)	λ_fl_ (nm)	ϕ_fl_	τ (ns)
H_2_O	329.0 ^D^/449.5 ^M^	618.5 ^a^	<0.01	<0.01
MgAPO-CHA	343.0 ^D^/437.0 ^M^	583.0 ^a^	0.33	2.73
MgAPO-AEL-1	*	610.0 ^a^	0.10	0.13 (20%) 3.07 (80%)
MgAPO-AEL-6	*	613.0 ^a^	0.41	3.30

* Absorption band too broad (deconvolution not possible, Appendix A). ^a^ Independent values regardless of excitation wavelength. D and M superscripts indicate dicationic and monocationic-species of 4-DASPI dye.

## Data Availability

Data is contained within the article and Appendix A.

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
