# Peer review of "Confinement of a Styryl Dye into Nanoporous Aluminophosphates: Channels vs. Cavities"

_ijms, 2024, doi:10.3390/ijms25073577_

Round 1

Reviewer 1 Report

Comments and Suggestions for Authors

Manuscript ID: ijms-2890201; Title: Confinement of a styryl dye into nanoporous aluminophosphates: channels vs cavities. In this paper, the author showcased two Mg-aluminophosphate zeotypes that were selected for the encapsulation of 4-DASPI dye (trans-4-[4-(Dimethylamino)styryl]-1-methylpyridinium iodide). By employing various magnesium-doped aluminophosphates (AEL and CHA) to provide a rigid environment for hosting 4-DASPI dye molecules, fluorescence quantum yields, and excited-state lifetimes have been significantly increased by slowing down the torsional motion of the dye. The resultant hybrid systems display significantly improved photophysical features compared to 4-DASPI in solution.

The topic is interesting for the referee. Therefore, I recommend publication after minor revisions.

1) I think that the abstract is very important for any paper. When I read the abstract, I didn’t have a good impression, and does not look effective. The author needs to rewrite the abstract and needs to include the motivation and necessity of this research.

2) I think that the introduction is not sufficient, and the introduction should be elaborated with state of art. Needs to include an introduction to are Advantages and Disadvantages.

3) Why the author has chosen styryl dyes for the research, the author needs to include in the manuscript what is the motivation of this research.

4) In Figure 1; the author needs to show D-π-A; which unit is donor and accepter for better understanding.

5) On Page No:5 and Table 2; the author has written “As expected, the dye shows poor fluorescence features in solution (ϕfl < 0.01 and τ < 0.1)”. what is the standard reference used for ϕfl calculations, I think that better to include.  Also, the author has written, “the fluorescence signal of particles did not show any intensity dependence, indicating that 4-DASPI molecules are incorporated inside the cages without any preferential orientation”. The author needs to explain detailed investigations.

6) Figure 5C; what is the excitation wavelength and monitoring emission wavelength for the lifetime, need to be included.

 7) On page No:2; the author explained previous work and wrote “reduction of the pore size from 10-rings as in AEL to 8-rings (small pore) might prevent the occlusion of these styryl dyes”. However, in the results and discussion, I didn’t find a detailed comparison study of the present work with previous work. In the result and discussion, the author needs to include a comparison of present work with previous work.

8) Please verify each sentence of this paper, Some typo errors.

Comments on the Quality of English Language

 Minor editing of English language required

Author Response

First and foremost, we would like to express our gratitude to the reviewer for finding our work interesting and for all his/her comments and suggestions, which will undoubtedly contribute to enhancing the quality of this work. In the following paragraphs, we provide a point-by-point response to the reviewer’ comments, detailing all the changes made in this revised version of our manuscript. The chances are highlighted in green.

Reviewer ·#1

Manuscript ID: ijms-2890201; Title: Confinement of a styryl dye into nanoporous aluminophosphates: channels vs cavities. In this paper, the author showcased two Mg-aluminophosphate zeotypes that were selected for the encapsulation of 4-DASPI dye (trans-4-[4-(Dimethylamino)styryl]-1-methylpyridinium iodide). By employing various magnesium-doped aluminophosphates (AEL and CHA) to provide a rigid environment for hosting 4-DASPI dye molecules, fluorescence quantum yields, and excited-state lifetimes have been significantly increased by slowing down the torsional motion of the dye. The resultant hybrid systems display significantly improved photophysical features compared to 4-DASPI in solution.

The topic is interesting for the referee. Therefore, I recommend publication after minor revisions.

1) I think that the abstract is very important for any paper. When I read the abstract, I didn’t have a good impression, and does not look effective. The author needs to rewrite the abstract and needs to include the motivation and necessity of this research; and 2) I think that the introduction is not sufficient, and the introduction should be elaborated with state of art. Needs to include an introduction to are Advantages and Disadvantages.

Authors reply: According to the reviewer's suggestions, we have modified the abstract and introduction sections, trying to clarify the main message of the work.

3) Why the author has chosen styryl dyes for the research, the author needs to include in the manuscript what is the motivation of this research.

Authors reply: The motivation for the tight confinement of styryl dyes into rigid zeolitic structures is deeply described at the beginning of the introduction in which we define styryl dyes, highlighting the advantages and disadvantages of their photophysical features.

To overcome their intrinsic poor emission efficiency, we propose the encapsulation in rigid environments in order to restrict their molecular motions to enhance their fluorescence. 

The innovation of this work is also highlighted in the introduction. We explained that we have previously worked with a styryl dye: LDS 722, for which occlusion within the AEL framework with 10-ring elliptical channels of 6.5 x 4 has boosted its fluorescence properties. In the present manuscript, we explore also a new type of framework, CHA, a three-dimensional system, with cages accessible through smaller 8-membered ring windows of 3.8 Å diameter, opening into large 6.7 Å x 10.0 Å ellipsoidal cavities that according to simulations could be optimal for confining 4-DASPI styryl dye. The present work analyzes the effect of this new type of confinement within zeolite cavities with respect to the one-dimensional channels of the AEL structure on the photophysical properties of 4-DASPI dye. This objective is detailed along the abstract and introduction and it is analyzed in the results and emphasized in conclusions.

4) In Figure 1; the author needs to show D-π-A; which unit is donor and accepter for better understanding.

Authors reply: The donor, acceptor and π-connector is now indicated in the molecular structure of 4-DASPI in Figure 1 and the caption.

5) On Page No:5 and Table 2; the author has written “As expected, the dye shows poor fluorescence features in solution (ϕfl < 0.01 and τ < 0.1)”. what is the standard reference used for ϕfl calculations, I think that better to include.

Authors reply: We are not able to record such very low quantum yields or short lifetimes of 4-DASPI dye in solution (they lie under the resolution of our spectrofluorometer). The very poor fluorescence of 4-DASPI is a very well-documented fact. Consequently, we have now included a new reference in the text (J. Phys. Chem. B 2008, 112, 7, 1906–1912) detailing the fluorescence quantum yield and the lifetimes of 4-DASPI in different solvents. We have also completed the information about the analysis of the lifetimes for the powder samples (page 11).

Also, the author has written, “the fluorescence signal of particles did not show any intensity dependence, indicating that 4-DASPI molecules are incorporated inside the cages without any preferential orientation”. The author needs to explain detailed investigations.

Authors reply: A further explanation of the isotropic behavior of 4-DASPI within CHA is now detailed in the main text (page 5)

6) Figure 5C; what is the excitation wavelength and monitoring emission wavelength for the lifetime, need to be included.

Authors reply: we thank the reviewer for the comment, the excitation wavelength is now included in the caption of Figure 5C.

 7) On page No:2; the author explained previous work and wrote “reduction of the pore size from 10-rings as in AEL to 8-rings (small pore) might prevent the occlusion of these styryl dyes”. However, in the results and discussion, I didn’t find a detailed comparison study of the present work with previous work. In the result and discussion, the author needs to include a comparison of present work with previous work.

Authors reply: we thank the reviewer for the suggestion, we have added the information and comparison of the former results centered on the encapsulation of another sytyl dye, LDS 722, into AEL in the results and discussion section (page 8)

8) Please verify each sentence of this paper, Some typo errors.

Authors reply: A thorough revision has been made.

Reviewer 2 Report

Comments and Suggestions for Authors

Oliden-Sánchez et al., submitted the manuscript entitled “Confinement of a styryl dye into nanoporous aluminophosphates: channels vs cavities”, to be published in “International Journal of Molecular Sciences (I.F = 5.6)”. This work can be accepted after addressing the queries.

  1. Introduction: Justify the need of this research with possible perspective applications.   
  2. Supplement HR-SEM, HR-TEM and AFM images of MgAPO-34 (CHA) framework and MgAPO-11 (AEL) framework to clarify the porosity and cavity.
  3. Explanation for XRD data must be boosted by stating the 2-Theta peaks and incredible findings with difference. Currently, this section looks very normal one.
  4. Supplement the separate table for fluorescent decay profiles (sating longer, shorter and average decay constants and components) with proper explanation.
  5. State the equation used to calculate the quantum yield in the materials and method section.
  6. Deliver the information of possible future direction in the conclusion section.
  7. Replace the old literature by recent relevant references.
Comments on the Quality of English Language

Minor English corrections are required

Author Response

First and foremost, we would like to express our gratitude to the reviewer for finding our work interesting and for all his/hercomments and suggestions, which will undoubtedly contribute to enhancing the quality of this work. In the following paragraphs, we provide a point-by-point response to the reviewer’ comments, detailing all the changes made in this revised version of our manuscript. The chances are highlighted in green.

Reviewer ·#2

Oliden-Sánchez et al., submitted the manuscript entitled “Confinement of a styryl dye into nanoporous aluminophosphates: channels vs cavities”, to be published in “International Journal of Molecular Sciences (I.F = 5.6)”. This work can be accepted after addressing the queries.

  • Introduction: Justify the need of this research with possible perspective applications.

Authors reply: according also to the suggestion of reviewer #1 (points 1-3) we have rewritten the abstract and introduction section of the manuscript to clarify the main goal of the work. The potential application of the present hybrid dye@aluminophosphate systems are mentioned in the results section (at the end of section 2.E) after the photophysical analysis of the samples (page 8).

  • Supplement HR-SEM, HR-TEM and AFM images of MgAPO-34 (CHA) framework and MgAPO-11 (AEL) framework to clarify the porosity and cavity.

Authors reply: SEM images were included in the original manuscript. With respect to TEM and AFM images, we consider they are not relevant for this work since the crystal morphology as well as the pore systems are well known for this type of zeolite materials, and such TEM and AFM analyses would not provide important information for the type of studies shown here and are well beyond the scope of our manuscript.

  • Explanation for XRD data must be boosted by stating the 2-Theta peaks and incredible findings with difference. Currently, this section looks very normal one.

Authors reply: XRD data are compared with the theoretical profiles provided by the database of zeolite structures of the International Zeolite Association (https://europe.iza-structure.org/IZA-SC/ftc_table.php), demonstrating beyond any doubt the crystallization of the proposed frameworks (CHA and AEL); this information has been included in the manuscript. This is done on a routine basis in zeolite research, it is the common way to show the crystallization of zeolite frameworks, and hence we consider that no further explanation is required.

  • Supplement the separate table for fluorescent decay profiles (sating longer, shorter and average decay constants and components) with proper explanation.

Authors reply: A complete description of the measurements and analysis of the decay lifetime curves are now added in the experimental section. The results are indicated in table 2. The goodness of the results is verified by the chi-square c2 values, which were always between 0.9 and 1.3. The explanation of a bi-exponential behavior was already hypothesized in the text (page 8) and was likely assigned to the coexistence of two phases, AEL and AFI, leading to a longer and shorter lifetime values in agreement with a more and less rigid confinement, respectively.

  • State the equation used to calculate the quantum yield in the materials and method section.

Authors reply: This information is added in the experimental section in the revised version.

  • Deliver the information of possible future direction in the conclusion section.

Authors reply: this information was already given in our original version of the manuscript, and the end of section 2.E.

  • Replace the old literature by recent relevant references.

Authors reply: We selected the references that we consider more relevant according to the topic of the manuscript (i.e. synthesis of AlPOs, photophysics of styryl dyes, …). We would like to ask the reviewer to be more specific and inform us about which significant bibliographic citations is missed in our work or should be replaced.

Round 2

Reviewer 2 Report

Comments and Suggestions for Authors

I am satisfied with author's reply. This paper can be accepted in its current version